# Dripplons as localized and superfast ripples of water confined between graphene sheets

Hiroaki Yoshida [1,2], Vojtěch Kaiser [1], Benjamin Rotenberg [3] & Lydéric Bocquet[1]

Carbon materials have unveiled outstanding properties as membranes for water transport, both in 1D carbon nanotube and between 2D graphene layers. In the ultimate confinement, water properties however strongly deviate from the continuum, showing exotic properties with numerous counterparts in fields ranging from nanotribology to biology. Here, by means of molecular dynamics, we show a self-organized inhomogeneous structure of water confined between graphene sheets, whereby the very strong localization of water defeats the energy cost for bending the graphene sheets. This leads to a two-dimensional water droplet accompanied by localized graphene ripples, which we call "dripplon." Additional osmotic effects originating in dissolved impurities are shown to further stabilize the dripplon. Our analysis also reveals a counterintuitive superfast dynamics of the dripplons, comparable to that of individual water molecules. They move like a (nano-) ruck in a rug, with water molecules and carbon atoms exchanging rapidly across the dripplon interface.

[1] Laboratoire de Physique Statistique, Ecole Normale Supérieure, UMR CNRS 8550, PSL Research University, 24 rue Lhomond, 75005 Paris, France. [2] Toyota Central R&D Labs., Inc., Nagakute, Aichi 480-1192, Japan. [3] Sorbonne Université, CNRS, Physicochimie des électrolytes et nanosystèmes interfaciaux, UMR PHENIX, F-75005 Paris, France. Correspondence and requests for materials should be addressed to L.B. (email: lyderic.bocquet@lps.ens.fr)

Nano-confined water appears in a wide variety of fields such as biology, geology, or tribology, and has long attracted significant attentions[1–3]. The recent emergence of two-dimensional materials such as graphene and graphene oxide (GO) membranes now offers the possibility to explore fundamentally the properties of water down to the molecular scale[4–13], together with direct applications in the fields of energy conversion, water desalination, dehydration or proton conduction in fuel cells[10,14–20]. Water confined in such 2D materials behaves quite differently from bulk water, leading e.g., to highly ordered structures even at room temperature, with lattice structures never observed in bulk ice, as revealed by means of both simulations and experiments[21–26]. Furthermore, peculiar dynamical properties have been observed for water flowing through nanoporous membranes, such as a fast transport and significant selectivity of solutions[27–30]. Understanding the behavior of confined water is thus crucial in order to control and optimize the performance of such new materials.

Water between two parallel solid plates is known to form ordered phases such as mono- and bi-layers, depending on the interlayer distance between the solid plates, as evidenced from various experiments e.g., in clays[31–34], between mica and glass[35], mica and graphene[36], graphene layers[23], and in GO[37]. In the graphene cases, elasto-capillary couplings of water phases with the confining substrate are not obviously expected owing to the high bending rigidity of the surfaces, graphene having a high bending modulus in the range of $B \sim eV \approx 40k_B T$ at room temperature. One may still define an elasto-capillary length as $\ell = \sqrt{B/\Delta\gamma}$, by comparing bending to a typical surface energy $\Delta\gamma$[38]. One obtains $\ell \sim 1$ nm (with $\Delta\gamma \approx 0.07$ Jm$^{-2}$ to fix orders of magnitude). This suggests indeed that water confined at sub-nanometer scales may actually couple to the graphene elasticity via surface energy.

In the present study, we investigate a peculiar inhomogeneity caused by the coupling mentioned above in a two-dimensional water thin film confined between graphene sheets, revealed using molecular dynamics (MD) simulations. A self-organized structure of water arises from the balance between the layering tendency of water and the stiffness of the graphene sheets. An example is shown in Fig. 1 (see also Supplementary Movies 1 and 2). Specifically, when the density of water molecules contained between the graphene sheets exceeds a threshold, a two-dimensional droplet of localized water is formed by bending the graphene sheets—here we call this droplet "dripplon." It has recently been shown that an isolated water droplet maintains its localized structures between two graphene sheets against the elasticity[39]. Here, we demonstrate the localization of water in water caused by the disjoining contributions, which is further relevant for lamellar

GO-like structures. We propose below a detailed understanding for the creation of this structure, coupling calculation of the free energy and disjoining pressure of mono- and bi-layers of water, together with a thermodynamic elasto-capillary modelization. The fast dynamics of the dripplon will be finally highlighted. Overall, the detailed implementation of the MD simulations is described in Supplementary Note 1. We quote that several water and graphene models have been tested, allowing to assess the robustness of our results.

## Results

**Water confined between rigid graphene sheets**. We start by considering the behavior of water between two rigid graphene sheets. In these first MD simulations, the graphene sheets are planar and constrained to move only in the perpendicular direction under a fixed pressure of 1 atm, whereas the fluid is maintained at a temperature of 300 K. The lateral size of the sheets shown in Fig. 2 is $S = 3.9 \times 4.3$ nm$^2$, but the behavior is essentially the same for larger sheets within the range we checked, up to $S = 15.7 \times 17.0$ nm$^2$ (see Supplementary Note 1.) In Fig. 2a, the interlayer distance $h$ between the two rigid graphene sheets in steady state is plotted as a function of the surface number density $\rho$ of water molecules (number of water molecules per unit area). Although $h$ increases almost linearly for high surface density, a plateau of $h \approx 0.95$ nm is observed in the range $\rho < 24$ nm$^{-2}$. In the latter case an ordered structure with a water bi-layer is formed. For $\rho \lesssim 13$ nm$^{-2}$, another plateau with $h \approx 0.67$ nm corresponds to a water monolayer. In this range, both the mono- and bi-layer are observed depending on the initial configuration, suggesting that both states are metastable under these conditions. The bi-layer at low density is associated with the nucleation of a liquid-vapor interface, the density of the liquid-like region remaining almost unchanged. Finally, a third state ($h \approx 0.8$ nm) is also observed in the narrow range of $13$ nm$^{-2} < \rho < 14$ nm$^{-2}$. Apart from their coexistence, water mono- and bi-layers formed between solid plates have widely been studied experimentally, e.g., in the crystalline swelling regime of clays[31–34] or the direct force measurement between surfaces[40,41]. The step $\sim 0.28$ nm observed here for the thickness of the water layer is in agreement with those experimental results.

The in-plane structure of the water mono and bi-layers is generally liquid-like, as observed e.g., in clays. Only for the monolayer with the highest density (at $\rho = 13.1$ nm$^{-2}$, on the lower plateau in Fig. 2a), close to the onset of coexistence, do we observe a square lattice with lattice parameter $\sim 2.8$ Å resembling the square-ice-like structure reported in ref.[23]. However, such a long-range order disappears if the confining graphene sheets are

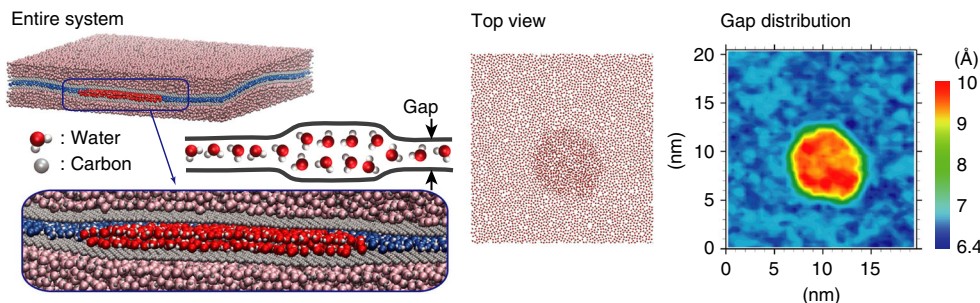

**Fig. 1** Simulated system, and a schematic representation of the two-dimensional droplet "dripplon". The two flexible graphene sheets with water molecules in between are sandwiched by two water reservoirs (fixing a hydrostatic pressure of 1 atm). The top view of the confined water molecules is also shown, along with the corresponding figure showing the distribution of the gap between the two graphene sheets. The water molecules in the reservoirs are light colored to focus on the confined water. The water molecules belonging to the dripplon are color-highlighted in red, whereas the confined water molecules outside the dripplon are blue-colored

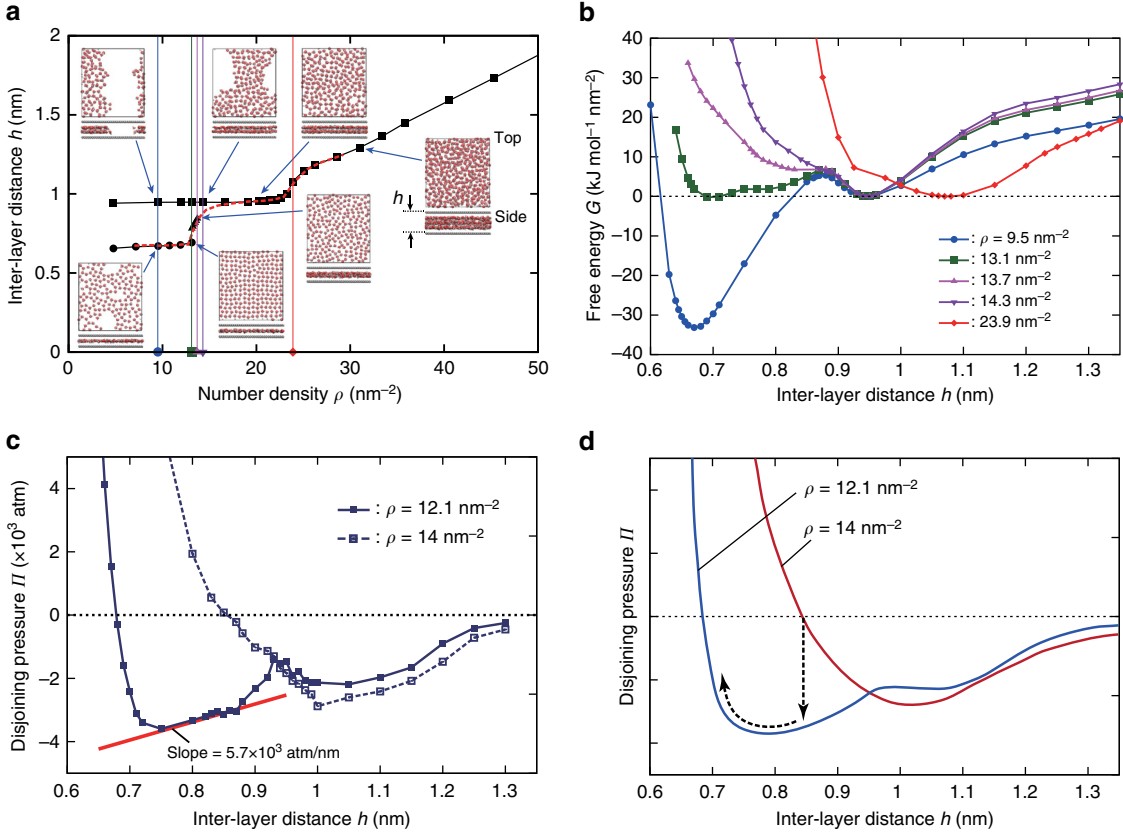

**Fig. 2** Properties of water confined between two rigid graphene sheets. **a** Interlayer distance $h$ versus number density per unit area $\rho$, obtained for the system of water molecules between two rigid graphene sheets of size $3.9 \times 4.3$ nm$^2$ under a pressure of 1 atm. The top views of the water configuration and the side views of the water and rigid graphene sheets at several points are presented inside the graph. The red-dashed line shows the result obtained for the small size system ($1.2 \times 1.3$ nm$^2$.) **b** Free energy per unit area $G$ at several values of $\rho$, along the vertical lines shown in **a**. The reference energy for each value of $\rho$ is chosen such that $G = 0$ at the local minimum in $h > 0.9$ nm. **c** Disjoining pressure $\Pi$ vs interlayer distance $h$ for the small size system ($1.2 \times 1.3$ nm$^2$), at $\rho = 12.1$ and 14 nm$^{-2}$. The red line shows the slope of $5.7 \times 10^3$ atm nm$^{-1}$ used in the analysis of spinodal decomposition process as schematically illustrated in **d**

flexible, as considered in the following sections, and only the local ordering may survive.

To confirm the coexistence of multiple stable states at low density, we show the free energy landscape in Fig. 2b. The free energy per unit area $G$ is plotted as a function of $h$, for the five values of $\rho$ indicated by the vertical lines in Fig. 2a. For each density, we performed series of MD simulations with fixed interlayer distances $h$ during which the pressure $p(h)$ was computed. The (swelling) free energy is then obtained using $G(h) = p_0 h - \int_{h_0}^{h} p(h')dh'$. Here, $p_0 = 1$ atm and $h_0$ is arbitrarily chosen such that $G = 0$ at the local minimum at $h > 0.9$ nm. Overall, the diagram in Fig. 2a is consistently explained by the free energy landscape. Specifically, there are two local minima for $\rho = 9.5$ nm$^{-2}$ corresponding to the mono- and bi-layer states, whereas only one local minimum at larger distance is found for $\rho = 23.9$ nm$^{-2}$. For $\rho = 13.4$ nm$^{-2}$ and 13.7 nm$^{-2}$, a shallow well corresponds to the state with $h \approx 0.8$ nm.

The nucleation of a liquid-vapor interface under some conditions, despite the entailed energetic cost, points to the high cohesive energy of water. The strong localization tendency of water results from its interaction with itself rather than with the confining walls and is also at the origin of the hydrophobic behavior of surfaces[42]. In the present case of extreme confinement down to the molecular scale, the instability of intermediate hydration states is also owing to the packing of discrete water molecules. The overall driving force for phase separation between mixed hydration states can be estimated in our molecular

simulations by considering systems too small to accommodate two phases separated by an interface, which therefore remain homogeneous. The interlayer distance obtained for a system of $S = 1.2 \times 1.3$ nm$^2$ is shown in Fig. 2a (red-dashed line). A single hydration state is observed, whatever the initial value of the interlayer distance. More quantitatively, the free energy $G$ for two densities is measured, to compute the disjoining pressure $\Pi = -\partial G/\partial h$, as plotted in Fig. 2c. The states are always unstable for the negative disjoining pressure ($\Pi < 0$). Furthermore, in the range where $\partial \Pi/\partial h > 0$, spinodal decomposition is expected once the homogeneity constraint is released. As we now proceed to show, the response of the system to this driving force for phase separation also depends on the flexibility of the graphene sheets, which results in a peculiar inhomogeneous interlayer structure.

**Graphene flexibility and localized dripplons**. We now relax the constraint of rigid graphene sheets and investigate how the water-graphene structure evolves. As depicted in Fig. 1, the graphene sheets are sandwiched by water reservoirs, which maintain a hydrostatic pressure of 1 atm (using pistons consisting of rigid graphene sheets at the boundaries of the reservoirs). For the purpose of examining the evolution of the hypothetical homogeneous state, let us consider a situation with an initial homogeneous state at $h = 0.85$ nm, which is the equilibrium state for an average number density $\rho_{av} = 14$ nm$^{-2}$ (Fig. 2c). If the density drops to $\rho_{av} = 12.1$ nm$^{-2}$, the system becomes unstable and the

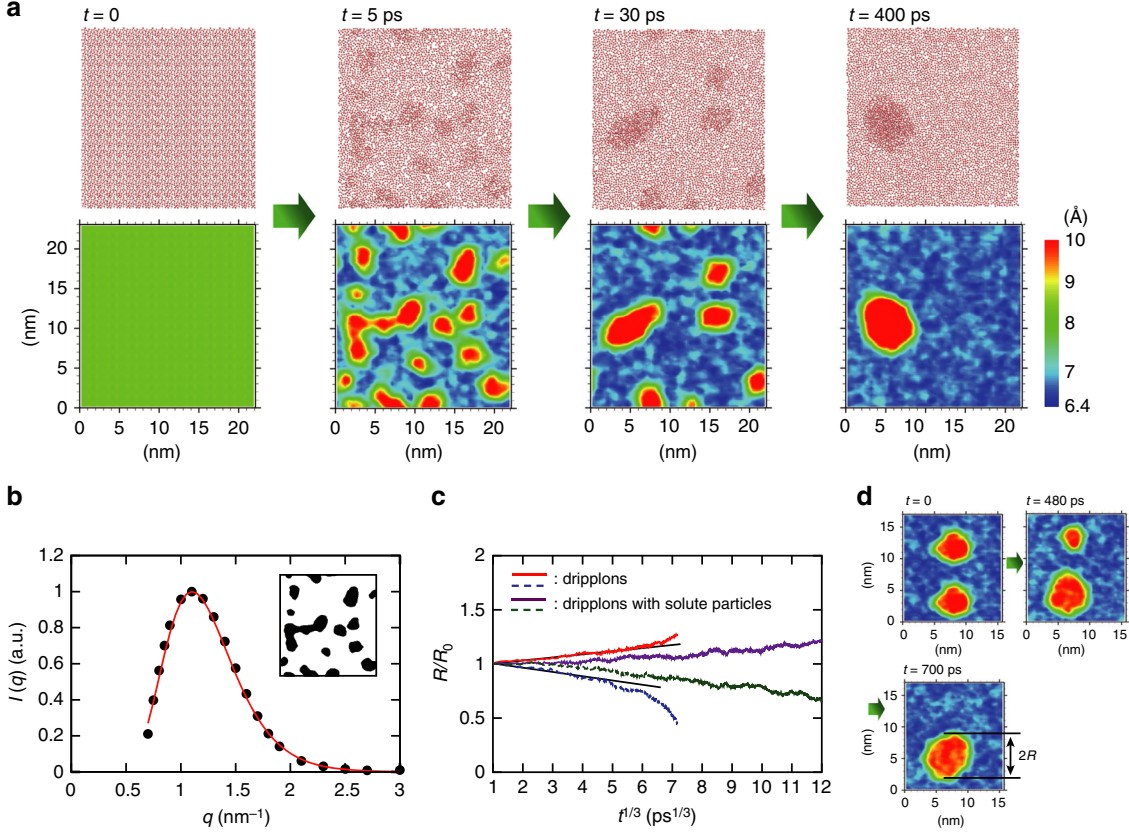

**Fig. 3** Spinodal decomposition and osmotic stabilization of dripplons. **a** Time evolution of the top views of the water molecules between flexible graphene sheets, starting from the homogeneous state at $h = 0.85$ nm. Each panel is accompanied by the distributions of gap between upper and lower flexible graphene sheets. See also Supplementary Movies 1 and 2. **b** Radially averaged scattering intensity function of the binary image (inset) of dripplons at $t = 5$ ps; five images from independent simulations are used for better statistics. The red curve shows a fit with a log-normal distribution. **c** Time evolution of initially two identical dripplons; the radius $R$, normalized by the initial radius $R_0$, is plotted as a function of $t^{1/3}$. The corresponding gap distributions of initially two identical dripplons are shown in **d**. See Supplementary Movie 3 for the video

phase separation takes place by spinodal decomposition, as illustrated in terms of the disjoining pressure in Fig. 2d. Although such a sudden decrease in the density is artificial, it allows to quantitatively investigate this instability which may occur during the drying process of e.g., a GO membrane.

Figure 3a shows snapshots illustrating the evolution of the initially homogeneous state at $h = 0.85$ nm, together with the distributions of the gap between carbon atoms of the upper and lower graphene sheets. (See Supplementary Movie 2 for the corresponding video). Here we clearly observe the instability and the phase separation into the monolayer water phase with $h \approx 0.67$ nm and the double-layer water phase with $h \approx 0.95$ nm. The resulting structure consists of a two-dimensional water droplet between the flexible graphene sheets, as illustrated in Fig. 1 (see also Supplementary Movie 1). Forming such a two-dimensional droplet requires overcoming the cost of the elastic energy to create the ripples in the graphene sheets and that of forming the interface between water mono- and bi-layers. We call the resulting combination of droplet and ripples a "dripplon."

The initial stage of the instability (Fig. 3a at $t = 5$ ps), understood as the spinodal decomposition discussed above, can be discussed more quantitatively. In Fig. 3b, we plot the scattering intensity as a function of the wave number, computed using the binary image of the interlayer distance distribution at $t = 5$ ps, as shown in the inset (we used five independent snapshots for better statistics, see Supplementary Note 2). The spectrum shows a log-normal distribution because the wave number has a lower bound

due to the size of the simulation box[43]. The fastest growing mode of the decomposition process is then obtained as $\lambda_m = 2\pi/q_m = 5.7$ nm, where $q_m$ is the wave number at the peak. We checked that this value was insensitive to the size of the simulation box.

This characteristic length for the spinodal structure can be estimated by extending the classical theory for the spinodal decomposition of thin films[44,45]. In the present case, the interfacial cost associated with the existence of a dripplon originates mainly from the elastic deformation of the graphene between the monolayer and bi-layer regions. Accordingly the surface energy cost is merely replaced by the elastic energy to bend the graphene sheets. Assuming a slight deformation of the film thickness, say $\delta h(x, y)$, the total free energy cost writes

$$\mathcal{G}[\delta h] = \int dS \left[ \frac{1}{2} B(\Delta \delta h)^2 + \frac{1}{2} \left[ \frac{\partial^2 G}{\partial h^2} \right]_0 \delta h^2 \right], \quad (1)$$

where $B$ is the bending modulus of graphene, $\Delta$ stands for the Laplacian and $\left[ \frac{\partial^2 G}{\partial h^2} \right]_0 \equiv -\left[ \frac{\partial \Pi}{\partial h} \right]_0$ (with $\Pi$ the disjoining pressure and $G$ the free energy per unit surface introduced above). Now, for a negative compressibility, i.e., $[\partial \Pi/\partial h]_0 > 0$, a prototypical spinodal scenario occurs and the fastest growing mode $\lambda_m$ is

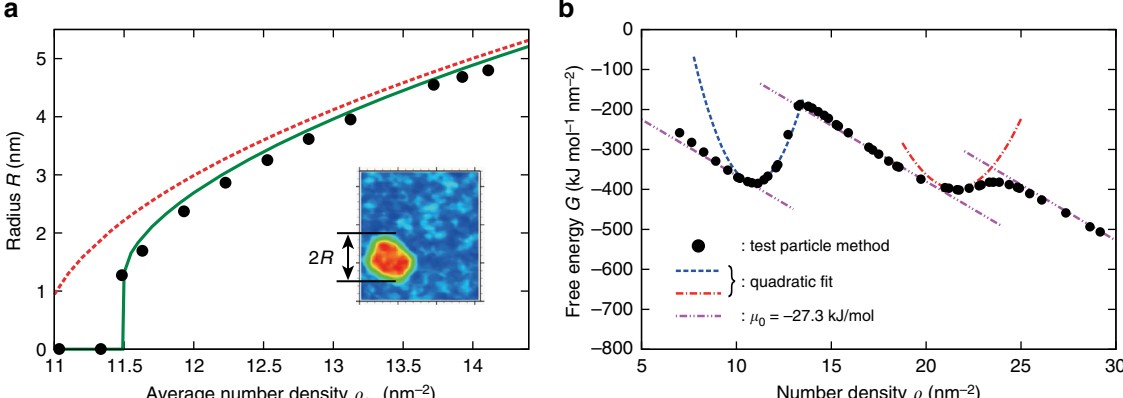

**Fig. 4** Thermodynamic model of dripplon creation. **a** Radius of the dripplon $R$ versus average number density $\rho_{av}$, for the system size of $15.7 \times 17.0$ nm$^2$. The solid line indicates the prediction of the thermodynamic model in Eq. (5). The prediction of the lever rule is shown by the dashed line. **b** Free energy per unit surface as a function of number density $\rho$. The symbols indicate the results obtained using the test particle insertion method. The dashed and dash-dotted curves are quadratic fits used in the thermodynamic model for the dripplon creation. The slope of the straight lines, $\partial G/\partial \rho = \mu_0 = -27.3$ kJ mol$^{-1}$, corresponds to the chemical potential of bulk water at 300 K and 1 atm

expressed as $\lambda_m = 2\pi\xi$, with

$$\xi = \left(\frac{B}{[\partial\Pi/\partial h]_0}\right)^{1/4}. \tag{2}$$

Numerically, using the values for the disjoining pressure provided in Fig. 2c, yielding $[\partial\Pi/\partial h]_0 \approx 5.7 \times 10^3$ atm nm$^{-1}$, we obtain the value of $\xi = 0.8$ nm for the fastest growing mode. This value is in excellent agreement with the value measured in the simulation images, yielding $\xi_{measured} = \lambda_m/2\pi = 0.9$ nm. This demonstrates that the driving force associated with water layering (disjoining pressure) is able to balance the considerable cost of interfacial energies originating in graphene deformation. This is a key for the dripplon existence.

**Ostwald ripening and dripplon osmotic stabilization.** After the initial decomposition process, the number of dripplons gradually decreases, until a single dripplon subsists (Fig. 3a, $t = 30 \sim 400$ ps). In order to examine this long-time behavior of multiple dripplons, we show the evolution of dripplon radii in Fig. 3c starting from a state having two identical dripplons in the same plane (prepared by replicating a smaller system with a single dripplon). The corresponding gap distributions are also shown in Fig. 3d (and in Supplementary Movie 3.) One of the dripplons shrinks over time, whereas the other grows until a single dripplon survives, without direct collisions. This process is interpreted as a two-dimensional version of Ostwald ripening[46,47]. The theory of Ostwald ripening predicts that the average radius of the drop evolves as $R(t) \sim t^{1/3}$. The plot of the radius as a function of $t^{1/3}$ does display a linear dependency, apart from the very final stage of vanishing the smaller dripplon. This implies that the process is indeed driven by the minimization of the interfacial energy, which arises at the perimeter of the dripplon.

However this ripening can be counteracted and dripplons can be stabilized at equilibrium. It is indeed well known that dispersed (nano-)droplets can be stabilized against Ostwald ripening process if they contain trapped species that are insoluble in the major phase[48,49]. Similarly here, adding a solute that is only stable in the water double-layer is accordingly expected to stabilize the dripplon. An osmotic pressure builds between the inner dripplon (with double water layers) and the outer region (with a single water layer), and it compensates at

equilibrium the interfacial pressure associated with the dripplon interface between the mono- to the bi-layer of water. The latter is defined in terms of a two-dimensional line tension $\gamma$, which accounts for the elastic deformation of the graphene sheets at this dripplon interface. A rough estimate of the corresponding elastic energy suggests that

$$\gamma \approx \frac{B}{\ell}, \tag{3}$$

with $\ell$ a molecular length. This yields typically $\gamma \approx 3 \sim 4 \times 10^{-10}$ J m$^{-1}$ (with $B = 2$ eV and $\ell \sim 1$ nm). The pressure balance thus leads to an estimate of the stable radius as

$$\frac{\gamma}{R_s} \approx \rho_s k_B T, \tag{4}$$

where $\rho_s$ is the solute surface concentration (see Supplementary Note 3 for details). Using $\rho_s = x_s \rho_2$ with $\rho_2 \simeq 20$ nm$^{-2}$ the density inside the dripplon and $x_s$ the molar fraction of solute— say, $x_s \sim 10\%$ to fix order of magnitudes—, one obtains an equilibrium radius of the dripplon of $R_s \approx 40$ nm. Such an equilibrium dripplon size is completely relevant experimentally but is currently out of reach for molecular simulations.

In the MD simulations, we examined however the effect of added solute particles in dripplons, using a feasible system size. As shown in Fig. 3c, we indeed observe as expected a slowdown of the coarsening ripening process. Here, we put three electrically neutral particles in each dripplon in Fig. 3c (see Supplementary Movie 3 for the corresponding video); the solutes are modeled with a Lennard-Jones interaction potential with a diameter comparable to two water molecules, so that they are less soluble in the water monolayer than in the bi-layer (see Supplementary Note 3). Such a finding not only confirms the mechanism at play for the formation of dripplons, but also suggests how such dripplon structures are expected to be stabilized by impurities in experiments.

**Thermodynamics of dripplon creation.** We now come back on the creation of the dripplon structure. As shown in Fig. 4a, the MD results demonstrate that there is a minimum water density for the creation of the dripplon. We thus explore further the thermodynamics of the dripplon and propose a model for this threshold mechanism.

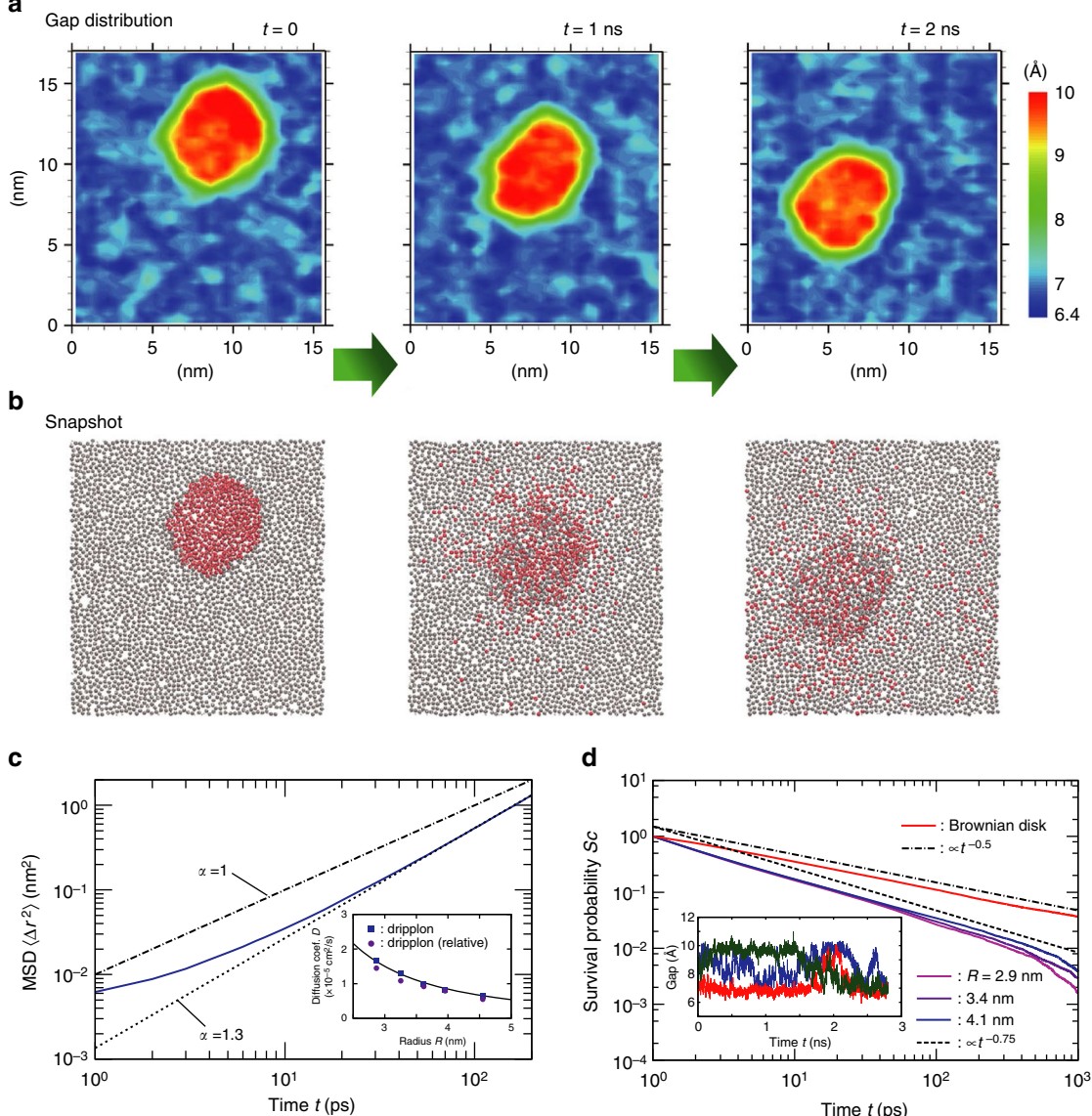

**Fig. 5** Dripplon dynamics. **a** Displacement of a dripplon over time, compared with diffusion of water molecules initially in the dripplon, which is visualized in **b**. The water molecules belonging to the dripplon at $t = 0$ are colored in red in **b**. The average density is $\rho_{av} = 12.3 \, \text{nm}^{-2}$, in the graphene sheets of $S = 15.7 \times 17.0 \, \text{nm}^2$. See Supplementary Movie 4 for the corresponding video. **c** Mean-square displacement (MSD) of a dripplon as a function of time, for a dripplon with $R = 2.9 \, \text{nm}$. The MSD highlights superdiffusive behavior, $\langle \Delta r^2 \rangle (t) \sim t^\alpha$ with $\alpha \approx 1.3$. The inset reports the calculated apparent diffusion coefficient of a dripplon (as defined in the main text). The solid line is a one-parameter fit as $cR^{-2}$. **d** Survival probability within the dripplon. $S_c(t)$ indicates the probability that a carbon atom participates in the dripplon for longer than time period $t$ (larger than 1 ps). The gap histories measured for three carbon atoms are shown in the inset

Let us consider a dripplon of radius $R$. The free energy cost associated with this structure can be written as

$$\mathcal{G}_{\text{total}} = \left(S - \pi R^2\right) G_1 + \pi R^2 G_2 + 2\pi R \gamma, \qquad (5)$$

where $S$ is the total area, and $G_1(\rho_1)$ and $G_2(\rho_2)$ are the free energies per area of the mono- and bi-layer phases, respectively. The last term is the free energy cost associated with the elastic bending of the graphene sheets at the dripplon perimeter. This is accounted for by a line tension $\gamma$, which relates to the bending modulus as in Eq. (4). We note here that a relevant theoretical approach using energy minimization has also been taken to explain the formation of bubbles on the surface of two-dimensional materials[50], although there are essential differences in main contributions to the total energy.

The free energy terms $G_1(\rho_1)$ and $G_2(\rho_2)$ are expected to exhibit minima associated with the mono- and bi- layer densities. This is evidenced in Fig. 4b, where we report the free energy calculated from the MD simulations. In practice, we have used Widom's test particle insertion method to compute the chemical potential $\mu(\rho)$ with the rigid graphene system[51,52], see Supplementary Note 4 for details of the test particle insertion method. The free energy is then obtained by thermodynamic integration $G(\rho) = \int_{\rho_0}^{\rho} \mu(\rho') d\rho'$. As shown in Fig. 4b, two local minima are clearly observed at $\rho_1 = 10.9 \, \text{nm}^{-2}$ and $\rho_2 = 21.5 \, \text{nm}^{-2}$, corresponding to water mono- and bi-layer states. The slopes of the linearly decreasing parts agree well with the reference chemical potential $\mu_0 = -27.3 \, \text{kJ} \, \text{mol}^{-1}$, which we obtained using the test particle simulation for a bulk water at 300 K and 1 atm, also in agreement with the literature[53]. These results confirm that the

free energies for the mono- and bi-layer states, $G_1(\rho_1)$ and $G_2(\rho_2)$, can be properly approximated by quadratic functions of the density: $G_i(\rho_i) = C_i(\rho_i - \theta_i)^2 + b\rho_i + d$. The parameters $C_i$, $\theta_i$, $b$, and $d$ are estimated from the MD free energy plot in Fig. 4b (see Supplementary Note 5).

Now using this expression for the free energy, we can calculate the equilibrium dripplon size $R$ for a given value of $\rho_{av}$ by minimizing $\mathcal{G}_{total}(\rho_1, \rho_2, R)$ in Eq. (5), under the constraint $(S - \pi R^2)\rho_1 + \pi R^2\rho_2 = \rho_{av}S$ (by means of the method of Lagrange multipliers). Figure 4a compares the theoretical prediction for the equilibrium dripplon radius as a function of average density with MD simulations results. A very good agreement is obtained, yielding a value $\gamma = 6.1 \times 10^{-10}\,\mathrm{J\,m^{-1}}$ for the line tension, adjusted to obtain a best fit. We quote that this value for $\gamma$ is in good agreement with the simple estimate suggested above, $\gamma \approx B/\ell$.

Overall, the thermodynamic model agrees very well with the simulation results. It predicts a minimal density for the creation of the dripplon, with the expression $\rho_c = \theta_1 + (3/2) \left[\pi\gamma^2 / \left(2C_1^2 S(\theta_2 - \theta_1)\right)\right]^{1/3}$. Using the parameters from the free energy, one predicts the value $\rho_c = 11.5\,\mathrm{nm^{-2}}$ for Fig. 4a, in very good agreement with the MD data. The prediction of the "lever rule", corresponding to $\gamma = 0$, fails to predict the observed threshold density. As a last comment, no upper bound on the dripplon size is expected. However on the simulation side, the size of the numerical box should exceed the expected dripplon size in order to avoid finite-size effects, see Supplementary Note 5.

**Superdiffusive dripplon dynamics.** The dynamics of the dripplon also highlights interesting features. In Fig. 5a, b (also in Supplementary Movie 4), we follow the motion of a dripplon and the underlying dynamics of water molecules belonging to the dripplon. A first striking remark is that the dripplons are fast moving objects, in spite of their extended nature (a dripplon with radius $R = 3$ nm contains typically 500 water molecules). This is highlighted in Fig. 5c where we report the mean-squared displacement (MSD) of a given dripplon versus time, as well as the extracted apparent diffusion coefficient. First the long-time behavior of the MSD is shown to be superdiffusive[54], with $\langle \Delta r^2 \rangle \sim \Gamma t^\alpha$. The exponent is typically $\alpha \approx 1.3$, substantially larger than unity. In the present case, one possible explanation for this superdiffusive behavior is the 2D nature of the dripplon dynamics, which is expected to enhance logarithmically the diffusion coefficient ($D(t) \sim \log t$ at long time) via the long-time relaxation of the hydrodynamic modes in 2D. However, it is interesting to note that the superdiffusive behavior of the extended dripplon contrasts with the bare diffusion behavior of individual water molecules, $\langle \Delta r^2 \rangle_{\mathrm{H_2O}} \sim t$, in spite of the 2D confining geometry (see Supplementary Note 6).

Even more striking is that the superdiffusive dynamics of the dripplon as an extended object is nearly as fast as individual water molecules, as highlighted by comparing the MSD, see Supplementary Note 6. If one defines an apparent diffusion coefficient of a dripplon, as $D_{app} = \Gamma t_0^{\alpha-1}/4$ with $t_0 = 0.2$ ns to fix ideas[55], then the values of $D_{app}$ are in the range of $10^{-5}\,\mathrm{cm^2\,s^{-1}}$. (We checked that the velocity autocorrelation function decays in the time scale of tens of ps, hereby supporting the choice of $t_0$.) These values compare with the self-diffusion coefficient of the bulk water ($\approx 2 \times 10^{-5}\,\mathrm{cm^2\,s^{-1}}$)[56]. This also compares with the diffusion of water molecules involved in dripplons ($D = 2.4 \times 10^{-5}\,\mathrm{cm^2\,s^{-1}}$ for water inside a dripplon—bi-layer—and $4.2 \times 10^{-5}\,\mathrm{cm^2\,s^{-1}}$ for water outside—monolayer–, in agreement with literature for confined water[57]). The motion of a dripplon with $D \sim 10^{-5}\,\mathrm{cm^2\,s^{-1}}$ thus appears as surprisingly fast for a collective object constituted of hundreds of molecules. In Fig. 5c-(inset), we also show the apparent diffusion coefficient of dripplons (defined above) versus their size. We find a decrease of the diffusion coefficient versus the dripplon radius, scaling typically as $\sim R^{-2}$, i.e., the inverse dripplon area. This is reminiscent of a similar observation for the motion of a droplet of water on free-standing rippled graphene[58], possibly suggesting that the energy dissipation is dominated by surface friction.

To get further insights into the coupled water-dripplon dynamics, we explored the relative motion of carbon sheets with respect to the water. First in Fig. 5b, we mark water molecules initially inside a dripplon in the snapshots and compare their diffusive dynamics to that of the dripplon shown in Fig. 5a. What emerges from this graph is that water molecules do not follow the dripplon, but are actively exchanged across the dripplon interface. In other words, the dripplon, defined as a mono- to bi-layer interface, constantly destroys and reforms itself while moving, just like a (nano-) "ruck in a rug"[59]: the motion of the dripplon structure is thus by nature collective.

Further support of this picture emerges from the survival probability $S_c(t)$ that a carbon atom belongs to the dripplon for more than a given time $t$, (out of those participating more than 1 ps). As shown in Fig. 5d, the probability exhibits a clear decay, confirming the local creation/destruction mechanism for every participating carbon atom. We compare the observed power decay, typically $\sim t^{-\beta}$ with $\beta \approx 0.75$, with the corresponding probability for a simplified model: namely the probability that a fixed point on a plane stays within a disk moving in the 2D plane under Brownian motion. This is a classical first-return problem, which leads to a survival probability decaying as $\sim t^{-1/2}$, as can be confirmed by means of a Brownian dynamics simulation of a disk (see Fig. 5d and Supplementary Note 6). The exponent $\beta \approx 0.75$ is slightly larger than the Brownian prediction, which again can be attributed to the superdiffusive nature of the dripplon as discussed above. Additional to the carbon motion, we also explored the water fluxes across the dripplon structure. To this end, we computed the probability $P_w(t)$ that a water molecule belongs to the dripplon for a time period $t$ (out of those inside the dripplon for more than 1 ps) and the corresponding survival probability $S_w(t) = 1 - \int_0^t P_w(s)\mathrm{d}s$. This estimate indicates that ~97% of water molecules inside a dripplon pass across the interface at least once within 100 ps, for all the dripplons up to $R = 4.6$ nm. This confirms a very active exchange of water molecules across the dripplon interface, whereas keeping the dripplon shape.

Altogether, these observations suggest that the dynamics of the dripplon on longer time scales, including its diffusion, reflects a collective behavior which goes beyond that of the constituting water molecules and carbon atoms. The dripplon is permanently destroyed and reformed, leading to fast motion.

## Discussion

We have shown that the mixed hydration states of confined water may couple to confining graphene flexibility to create localized water drops accompanied by a strong deformation of graphene sheets. The elastic bending energy cost creating the ripples is defeated by the strong preference of water layering to homogenization. The emergence of this localized structure is rationalized by means of a thermodynamic model, highlighting the competition between the disjoining pressure and surface elastic costs of the graphene sheets. This points to an interesting analogy between the behavior of molecularly confined water between graphene sheets, and that of thin liquid soap films. This is in contrast to the rigid layered materials with intercalating water, such as the swelling clays[33], which are too rigid to sustain such

localized defects thus rather favoring the interstratification of the confined water with different hydration states.

From an experimental point of view, such structures could manifest themselves in systems of layered graphene, such as with GO membranes. The condition for the creation of the dripplon requires to reach the threshold density $\rho_c$, which is expected to occur in the drying process of GO membranes. Also the typical lateral size of the sheets of the GO membrane is far larger (~ cm) than those considered in the present study[29,30] and hence the water density inside could be maintained in a quasi-stationary state under ambient conditions, favoring the apparition of dripplons. Let us also note that the molecular layering in the vicinity of the liquid–solid interface is quite general[60], and it is experimentally observed for various liquids, such as alcohols and hydrocarbons[61,62]. The physical picture of the dripplon can therefore be possibly applied to various liquids other than water, confined between graphene-based membranes.

Furthermore, we showed that solute particles and impurities contained in the dripplon lead to an osmotic stabilization of a dripplon, against the Ostwald ripening. This suggests the robustness of equilibrium dripplons in real systems. Reversely, this solute-induced stabilization leads to a strong 'wetting' of the solute particles within the dripplon: solutes and impurities are dressed in multi-layers of water and concentrated in these localized structure. This effect could have an important role in determining the structure of multilayer graphene-like GO when dried.

An interesting perspective concerns particle transport mediated by the dripplon dynamics. These fast moving ripples could indeed act, to some extent, as cargoes transporting confined particles between the sheets. Indeed, the fast dynamics of these extended objects could allow to boost the transport of particles across these layered materials by adding new degrees of freedom for the strongly confined solute, thereby enhancing its dynamics. This is somewhat reminiscent of the coupling of water dynamics to that of the (fluctuating) confining surfaces as shown recently in the context of carbon nanotubes[63]. This suggests to explore the interplay between the dynamics of the (fast) dripplon and that of (slow) suspended particles. From a more general perspective, these fast dripplon defects should impact water and ionic transport across such layered materials, as well as salt rejection. Such behavior opens new exciting leads, which remain to be explored, including e.g., the manipulation of dripplons and embedded solutes via applied electric fields or osmotic gradients.

We hope that our results will seed experimental investigations to search for these predicted exotic structures and their consequences on membrane transport.

**Data availability**. The data that support the findings of this study are available from the corresponding author on request.

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

## Acknowledgements

L.B. and B.R. acknowledge funding from the Agence Nationale de la Recherche (ANR), project "Neptune". This work was granted access to the HPC resources of MesoPSL financed by the Region Ile de France and the project Equip@Meso (reference ANR-10-EQPX-29-01) of the program Investissements d'Avenir supervised by ANR.

## Author contributions

L.B. conceived the project. H.Y. performed the simulations and H.Y., V.K., B.R., and L.B. analyzed the data. H.Y., B.R., and L.B. wrote the manuscript.

## Additional information

**Competing interests:** The authors declare no competing interests.

