## [Peer Review File · Nature Communications]

Reviewers' comments:

Reviewer #1 (Remarks to the Author):

In this manuscript, the authors study a self-organized inhomogeneous structure of water confined between graphene sheets, termed as driplon. The presence of driplon is rationalized using a thermodynamic model. The evolution of the structures is described in terms of spinodal decomposition process and Ostwald ripening. The kinetics of the driplon is discussed based on diffusion related process. The topics of the manuscript is interesting and the phenomenon shown is new. The referee can suggest acceptance after the following issues addressed by the authors.

1. The authors calculate the free energy for structures with different surface density of water molecules ρ as a function of interlayer distance (Fig.2b) h . It is not clear that how one can fix ρ while changing h . This should be explained clearly.
2. In Fig.5b the authors estimate the diffusion behavior of driplons by analyzing their MSD during a duration about 0.1 ns. For nanoclusters, this duration is usually not long enough as it has been shown that the diffusion can transform from power-law type to normal diffusion with MSD obtained during tens of ns (Phys. Rev. B, 67 (2003) 085406). Thus it is necessary to show the dependence of MSD on time in longer duration, e.g. 10 ns.
3. When calculating the long term behavior using MD simulation, one important issue is the convergence of the simulation. It would be more convincing by showing the energy drift of the simulation under NVE ensemble, e.g. how many eV per degree-of-freedom per ns.
4. In Fig.4a the authors show that there is a lower bound for the size of driplons. They also discuss the increase of the size in terms of Ostwald ripening. It would be interesting to know whether there is an upper bound for the size of the driplons assuming there is no solute, i.e. pure water.
5. On line 118, it seems that the unit for free energy is N/m instead of N·m, i.e. J. Is it correct?
6. On line 299, the word the is duplicated.

Reviewer #2 (Remarks to the Author):

This manuscript reports a detailed computational and theoretical analysis of the formation of ripples due to the confinement of water between two graphene sheets. The authors report molecular dynamics simulations of water sandwiched between two graphene sheets initially constraining the graphene sheets to maintain their 2D flat geometry and then relaxing the sheets and allowing them to bend. They analysed the bending of the sheets and the formation of ripples (that they called driplons and have been also called “bubbles” in other papers) using the physical chemical theories used to explain the formation of bubbles e.g. a competition between the (graphene) bending rigidity and the (water/graphene) surface energy.

The formation of ripples on graphene (but also other 2D materials) has been the subject of several recent high profile published papers and the topic is of interest from a basic science point of view but also for practical applications. The simulations reported have been properly performed and the analysis carried out to explain the observed bending of the graphene sheets is interesting. As said the manuscript covers a topic which has been quite extensively studied recently and for which experimental data are robust in terms of the formation of the ripples/bubbles but much less clear in terms of what type of structure the water forms within these enclosures.

Despite the interesting analysis and the large number of simulations performed, I do not find the manuscript suitable for the publication in nature communications in this form and I'd like the authors to reply to the following comments before considering its publication:

1. While the experimental data on the formation of the bubbles is solid less clear is what happens to the water within these enclosures. The experimental references that the authors used to back up some of their results about the formation of mono and bilayers are either disputed now in the literature see ref. 9 or do not provide any clear indications about the formation of crystal/solid structure inside nanoscopic enclosures of the type studied here (the references to the clays studies published in the 50' do not refer to these type of confinements). For these reasons the plots reported in Fig. 2 for examples are not backed up by any experimental data and could be the results of water models the authors have employed. The authors themselves mentioned in the SI that if another water model is used different results are obtained.

2. the calculations of the free energy G show that the single and double layer structures are thermodynamically stable for the chosen water and graphene models, but do not show that these structures really exist as they are, I suspect, water model dependent. Therefore how can we trust quantitatively the results?

3. The manuscript basically show that ripples formation is possible even though the graphene sheets have very high bending rigidity. The authors back up they MD results using surface physical chemistry and show that the ripples is driven by the interfacial energy. I'd like the authors to compare their analysis with that recently published in Nat. Comm.

<https://www.nature.com/articles/ncomms12587> where a similar theoretical analysis has been performed. Here I think lays my biggest concern: while I find the theoretical analysis interesting (although to me it looks similar to that published already in Nat Comm) I have doubts on the interpretation of the MD results and specifically their dependence on the employed water model which as I said the authors trust to be reliable in these confined situations but that has instead been parametrised to reproduce the structure of standard ice.

Reviewer #3 (Remarks to the Author):

Water between two parallel solid plates is known to form mono-, bi-layers or more layers, depending the water content between the solid plates. Specifically, once the density of water molecules exceeds the threshold of a monolayer, a local two-dimensional or more layered water will be formed. By employing the molecular dynamics simulation, the authors studied the inhomogeneous water thin film confined between graphene sheets. They found a self-organized structure of confined water and such a phenomenon is explained by the balance between the layering tendency of water and the stiffness of the graphene sheets. The simulation itself do not give much novelty [Sci Rep. 2017; 7: 2646; J. Am. Chem. Soc., 2008, 130 (6), pp 1871–1878; Chin. Phys. B Vol. 26, No. 10 (2017) 106401], whereas the thermodynamic explanation is interesting. I may recommend it for publication in Nature Communications if the authors can answer the following questions satisfactorily.

1. The shape of the bi-layer driplon (in Fig.3a $t=400$ ps) indicates that the 2D water is isotropic along in-plane directions. How to control the moving direction of the droplet in practical application?

2. A further problem, is there any influence of the anisotropic structure of graphene on the driplon? What's the influence? i.e., armchair direction or zigzag direction.

3. The fastest growing mode is estimated to have a wavelength of 5.7nm, what is the meaning of the equilibrium wavelength should be 5.7nm?

4. Why the diameter ($2R_s$) of the driplon at $t=400\text{ps}$ is larger than the equilibrium wavelength 5.7nm ?

5. In the main text, the authors sometimes call these extra water molecules defects, sometimes call them driplon. I cannot find the definition of the defect through the manuscript. Are these terms different?

6. The formation of monolayer or bilayer is described by phase separation process. As long as the water density along thickness direction is inhomogeneous, such a phenomenon (FIG. 3) should widely exist. Why does the phase separation disappear, as shown in FIG. 2, when the number density ρ is large enough? i.e., above 23.9nm^{-2} .

7. There are many mistakes in the manuscript. For example, the density in line 104 should be 13.1nm^{-2} instead of 1.31nm^{-2} . The authors should check the whole manuscript carefully.

8. The potential applications of the so-called driplon should be discussed, otherwise, there is not much difference from playing with boundary value problem by molecular dynamic simulation.

9. I cannot find the citation of the relevant work in the references [Sci Rep. 2017; 7: 2646; J. Am. Chem. Soc., 2008, 130 (6), pp 1871–1878; Chin. Phys. B Vol. 26, No. 10 (2017) 106401; J. Phys. Chem. C 2007, 111, 1709-1715, etc], the difference from previous work should be explained and emphasized.

Reply to the referees' comments

In the revised manuscript, all the main revisions are highlighted in **red font**. The following is our reply (in **blue font**) to the comments (in *black font*). The corresponding revisions made in the manuscript are stated in each reply (highlighted in **bold font**). The citation and equation numbers refer to those **in the revised manuscript**.

General note: Before entering our detailed answer to all raised questions –see below–, we would like first to thank all three referees for their positive feedback and fruitful comments. We would like to emphasize also that following referees' remarks, we have performed complementary (and exhaustive) simulations and thermodynamic analysis in order to fully assess our conclusions. These new data fully support the mechanisms underlying the driplons creation and the proposed thermodynamic framework accounting for it. This, we believe, has allowed to strengthen further our message, making our conclusions very robust. All in all, we think that the manuscript strongly benefited from these insights from the referees.

Answer to Referee #1

We are grateful to the referee for his/her thorough review of the manuscript and valuable comments. The replies to the comments are listed in the following.

■Comment 1.1

The authors calculate the free energy for structures with different surface density of water molecules ρ as a function of interlayer distance (Fig.2b) h . It is not clear that how one can fix ρ while changing h . This should be explained clearly.

●Reply 1.1

Here, ρ is the surface density, number of molecules per unit area. Since the area in the x - y plane does not change in the simulation, the value of ρ is an input parameter that we can control. To avoid this confusion, we added an explanation **“number of water molecules per unit area” on line 92, and “per unit area” in the caption of Fig. 2a.**

■Comment 1.2

In Fig.5b the authors estimate the diffusion behavior of driplons by analyzing their MSD during a duration about 0.1 ns. For nanoclusters, this duration is usually not long enough as it has been shown that the diffusion can transform from power-law type to normal diffusion with MSD obtained during tens of ns (Phys. Rev. B, 67 (2003) 085406). Thus it is necessary to show the dependence of MSD on time in longer duration, e.g. 10 ns.

●Reply 1.2

The question actually points to the relaxation time scale of the driplon system, in order to obtain proper time-dependent statistics. Below she show the results for the velocity autocorrelation function of the driplon motion, which allows to obtain the diffusion coefficient by integration: Figure R1a plots $\langle \mathbf{v}(t) \cdot \mathbf{v}(0) \rangle$ (normalized by $\langle \mathbf{v}(0)^2 \rangle$), as a function of time. As shown on this figure, the relaxation time-scale is in the range of pico-seconds. Accordingly, simulation aiming at calculating the MSD up to ~ 100 ps in duration allows to obtain a proper estimate of the diffusion coefficient. That being said, we agree with the referee that it is important to check the power-law dependence of MSD for longer time range. Accordingly we have performed additional very long simulations (over 10 ns, as suggested by the referee) for one case ($R = 2.9$ nm) to obtain smooth MSD for very long time range (longer than 1 ns). We quote that in the present situation obtaining MSD in long duration \sim ns is extremely demanding for long duration with good statistics (furthermore we perform several independent simulations to obtain a proper statistics for the driplon trajectories). As shown in Figure R1b, the long-range MSD confirms the power-law dependence, which safely support the discussion in the original manuscript. We have made a remark on this issue **in the revised manuscript around line 405.**

Figure R1: Velocity autocorrelation function $|\mathbf{v}(t) \cdot \mathbf{v}(0)/\mathbf{v}(0)^2|$ as a function of time, where \mathbf{v} is the velocity of the center of mass of a driplon (of $R \sim 2.9$ nm.) **a**: Mean-square displacement (MSD) of a driplon as a function of time, for the same driplon. The MSD shows super-diffusive behavior, $\langle \Delta r^2 \rangle(t) \sim t^\alpha$ with $\alpha \approx 1.3$.

■ **Comment 1.3**

When calculating the long term behavior using MD simulation, one important issue is the convergence of the simulation. It would be more convincing by showing the energy drift of the simulation under NVE ensemble, e.g. how many eV per degree-of-freedom per ns.

● **Reply 1.3**

We do not fully grasp the concern of the referee, *i.e.* what is meant by the “convergence” of the simulation (and the information provided by simulations in the alternative NVE ensemble). First we emphasize that we are using standard algorithms, with a very well established MD code, LAMMPS, which has been exhaustively benchmarked by thousands of simulations over 25 years of practice worldwide. The present configuration with water in carbon confinement has also been the object of numerous investigations in various forms.

Here we use the Nosé–Hoover thermostat to fix the temperature and we have checked that the temperature of water between the graphene sheets is properly controlled with the thermostat applied to the reservoirs. The temperature remains constant over the full simulation time. Furthermore – and this is maybe the most important here – we checked that after a required equilibration time, all equilibrium and thermodynamic quantities are stationary, and no drift is measured. For example the disjoining pressure of the slab, for various gaps; or the water diffusion coefficient which has properly converged to expected values; or the size of the driplon which (once generated) remain stable and constant over the whole simulation times. We never measured any sign of drift or absence of convergence in any of the (many) simulations which we have performed. We have finally checked that the chosen time step is sufficiently small to ensure that the relevant quantities for the present case of fixed temperature and pressure are conserved during the whole duration of our simulations.

Also, and as is usual in simulations to explore phase change, we do fix the pressure in the simulations as well by using pistons in the reservoirs, **see Supp. Info. S1 for details**. Accordingly we cannot properly fix the energy in this configuration while having pistons. Now, avoiding pistons and fixing the total volume of the reservoirs (eg to perform a full NVE simulation) would require a fine tuning of the chosen volume in order to obtain the targeted pressure in each configuration. We do not feel that this is appropriate in the present case since it would make far more difficult the exploration of the phase change at stake here.

■ **Comment 1.4**

In Fig.4a the authors show that there is a lower bound for the size of driplons. They also discuss the increase of the size in terms of Ostwald ripening. It would be interesting to know whether there is an upper bound for the size of the driplons assuming there is no solute, *i.e.* pure water.

● **Reply 1.4**

This is indeed an interesting question. The thermodynamic model indeed shows that the radius of the driplons increases as $R \sim \sqrt{\rho_{av}}$ (cf. Eq. (22) of Supplemental Information). To our knowledge, there is no

obvious fundamental (physical) mechanism which can limit the size of the driplon and therefore no upper bound is expected. We have added several simulation results to Fig. 4a, with new data at $\rho_{av} = 13.9$ and 14.1 nm^{-2} , confirming further the size increase and the good agreement with the thermodynamic model.

However in the simulations, the finite lateral size of the box will constitute a *numerical* (but not physical) bound, due to the periodic boundary conditions. When the driplon diameter reaches the lateral box size, the driplon will “feel” itself. Such a situation is shown in the figure below, reporting additional simulations showing the outcome with density $\rho_{av} = 14.3$ where the predicted driplon diameter reaches the lateral box size L . As shown in Figure R2b, the driplon is shown to persist ... but its shape changes due to the periodic boundary conditions: it connects with itself and takes the form of a stripe of bilayer water. Such a change of shape is expected to minimize free energy in a finite box: the line energy of a circular driplon with diameter L (equal to the box size) is $\gamma \times \pi L$ to compare with two stripes of length L with an energy $\gamma \times 2L$. The latter being smaller (for any driplons with diameter above L), it is expected that the system changes its shape when the driplon size reaches the simulation box size L . But this is a numerical bound due to finite size effects and not a physical bound.

We now quote this point **in the revised version around line ~ 370** , and we mention these data **in Supplemental Information Sec. S5**.

Figure R2: **a**: Radius of the driplon R versus average number density ρ_{av} , for the system size of $15.7 \times 17.0 \text{ nm}^2$. The solid line indicates the prediction of the thermodynamic model in Eq. (5). The prediction of the lever rule is shown by the dashed line. **b**: Distributions of gap between upper and lower flexible graphene sheets at $\rho_{av} = 14.1$ and 14.3 nm^{-2} .

■ **Comment 1.5**

On line 118, it seems that the unit for free energy is N/m instead of $N \cdot m$, i.e. J . Is it correct?

● **Reply 1.5**

We thank the reviewer for pointing to this confusion. The unit of free energy is indeed N/m and the unit appearing in Figs. 2b and 4b was intended to signify N/m . In the revised manuscript, we added parentheses (i.e. $\text{kJ}/(\text{mol nm}^2)$) to avoid this confusion in those figures **2b** and **4b**.

■ **Comment 1.6**

On line 299, the word the is duplicated.

● **Reply 1.6**

Thanks, we corrected this and other minor typos.

Answer to Referee #2

We appreciate the thorough review of the manuscript by the referee and his/her valuable suggestions.

Overall the referee raises the question of the robustness and sensitivity of our results to the model of water used. We answer below these points in detail. But motivated by the referee's question, we performed a complete new set of simulations with different water models. Not only we could confirm the existence of driplons with these other models but we also reproduced the detailed thermodynamic analysis underlying their existence. Indeed one key ingredient of our approach is the existence and thermodynamic stability of single and double layer structures. This required massive complementary simulations with corresponding free energy calculations. **Altogether these new simulations results fully confirm the general picture and the framework proposed.** All in all, the formation of driplon is a robust result because it originates in the balance of robust ingredients: water layering combined elastic free energy cost. The fact that we recover the same results whatever the water model confirms this robustness.

These new results allay the concerns of the referee. We believe that they also strengthen considerably our results and improve the quality of the manuscript.

The replies to the comments are listed below.

■Comment 2.1

While the experimental data on the formation of the bubbles is solid less clear is what happens to the water within these enclosures. The experimental references that the authors used to back up some of their results about the formation of mono and bilayers are either disputed now in the literature see ref. 9 or do not provide any clear indications about the formation of crystal/solid structure inside nanoscopic enclosures of the type studied here (the references to the clays studies published in the 50's do not refer to these type of confinements). For these reasons the plots reported in Fig. 2 for examples are not backed up by any experimental data and could be the results of water models the authors have employed. The authors themselves mentioned in the SI that if another water model is used different results are obtained.

●Reply 2.1

The main message that we intended to deliver with Fig. 2 is that water preferably form the mono- and bi-layer structures when it is confined in a gap of ~ 1 nm, which has long been known in the field of clay swelling. The stepwise behavior in Fig. 2a with the step ~ 0.28 nm is indeed consistent with the experiments reported in Ref. 32 (Kraehenbuehl et al.) and with more recent experiment reported in **newly cited Ref. 34** (Dazas et al.). We indeed cited Ref. 9 in the discussion on the structure of the confined water to mention the *locally* structured water observed in our simulations, but we did not intend to refer is as an experimental backup.

However, we fully agree with the reviewers in that other water models should be examined. As stated in the introduction above, we performed additional simulations using SPC/E model and TIP4P/2005-Flexible model [J. Chem. Phys. 135, 224516 (2011)] to examine the effect of water model on the structuration. The results are shown in Fig. R3 below. All models agree completely on the phase behavior, *e.g.* interlayer distance versus number density, Fig.R3-a. But going beyond, we have performed an exhaustive analysis of the thermodynamics of the system using the various water models. This required a full calculation of the thermodynamics (calculating free energy by thermodynamic integration). While there are some minor quantitative changes - as expected since the models differ from each other -, all model fully agree on the existence of free energy minima for mono- and bi- layers. As we discuss in the paper, this is the main required ingredient for the existence of localized driplon structure along the thermodynamic framework. Finally we have also checked the existence of driplons for all these water models, as shown in Fig. R4. All together, the robustness of formation of driplon has been confirmed.

We have added this new figure as **Fig. S1 and S2 in the revised Supplemental Information.**

Figure R3: **a:** Inter-layer distance h versus number density ρ , obtained for the system of water molecules between two rigid graphene sheets of size $3.9 \times 4.3 \text{ nm}^2$ under a pressure of 1 atm, using different water models. **b-d:** Free energy per unit area G versus inter-layer distance h at several values of ρ . **b:** TIP4P/2005, **c** SPC/E, and **d** TIP4P/2005-Flexible water model. The reference energy for each value of ρ is chosen such that $G = 0$ at the local minimum in $h > 0.9 \text{ nm}$.

Figure R4: Gap distributions of the two graphene sheets along with the corresponding top views of the confined water molecules. **a** TIP4P/2005 water model, **b** SPC/E water model, and **c** TIP4P/2005-Flexible water model.

■Comment 2.2

The calculations of the free energy G show that the single and double layer structures are thermodynamically stable for the chosen water and graphene models, but do not show that these structures really exist as they are, I suspect, water model dependent. Therefore how can we trust quantitatively the results?

●Reply 2.2

As mentioned in the above Comment 2.1, we carried out computations to calculate the free energy G to show that the single- and double-layer structures are stable, using different water models, *i.e.* SPC/E and TIP4P/2005-models on top of the TIP4P/2005. The results are shown in Fig. R3c and d (**and Fig. S2 in the revised Supplemental Information.**) As pointed out by the referee, there are indeed slight quantitative differences between the results, but the most important point, *i.e.* the thermodynamic stability of single and double layer structures is fully confirmed for all water models. This confirms the robustness of the presented phenomenon.

■Comment 2.3

The manuscript basically show that ripples formation is possible even though the graphene sheets have very high bending rigidity. The authors back up they MD results using surface physical chemistry and show that the ripples is driven by the interfacial energy. I'd like the authors to compare their analysis with that recently published in Nat. Comm. <https://www.nature.com/articles/ncomms12587> where a similar theoretical analysis has been performed. Here I think lays my biggest concern: while I find the theoretical analysis interesting (although to me it looks similar to that published already in Nat Comm) I have doubts on the interpretation of the MD results and specifically their dependence on the employed water model which as I said the authors trust to be reliable in these confined situations but that has instead been parameterised to reproduce the structure of standard ice.

●Reply 2.3

The paper by Khestanova *et al.* [Nat. Comm. 2016]) is indeed interesting (unfortunately we missed this recent reference in the first place). In this paper Khestanova *et al.* show that the localized structure that occurs in two-dimensional materials is very important, and it has begun to attract much attention recently. The bubbles appearing in the two-dimensional materials are seemingly relevant to the driplons in our paper. We quote however that in the paper by Khestanova *et al.*, no water is present between the layers. Accordingly their theoretical framework does not include *disjoining pressure of water*, which is actually the crucial ingredient of the phenomenon at stake in our study: the disjoining pressure balances with the elastic contribution in creating driplons. It is interesting to actually highlight that localized ripples are maintained in spite of the presence of water. This is due to the layering effects originating in the disjoining pressure of water, as demonstrated by our thermodynamic model. Even more water layering favors localized structures, and – as quoted by the referee – is able to bend the graphene sheets.

We thus believe that our work nicely extends on the results of the paper by Khestanova *et al.* by showing that the presence of water – as occurs for wetted layered graphene materials in the context membranes – favors the occurrence of bubbles. This points to the importance of localized structures in graphene materials and should motivate experimentalists to search for these.

We cite it in the revised manuscript as Ref. 50 in the sentence around line 323.

Answer to Referee #3

We appreciate the thorough review of the manuscript by the referee and his/her valuable suggestions. This, we believe, improved the quality of the manuscript to a significant extent. The replies to the comments are listed below.

■Comment 3.1

The shape of the bi-layer driplon (in Fig.3a $t = 400ps$) indicates that the 2D water is isotropic along in-plane directions. How to control the moving direction of the droplet in practical application?

●Reply 3.1

This is an interesting question. As pointed out by the reviewer, in the present study, the static and dynamical properties of the driplon are isotropic. This is because the mechanical properties of the graphene sheet are almost isotropic as mentioned in Reply 3.2.

We could think of controlling the direction of the moving droplet by several actions: for example in the presence of dissolved salts and electric fields, one may expect to generate a specific axis for motion (with the direction possibly depending on the net charge engulfed in the driplon). Alternatively the action of temperature gradients would definitely have a consequence in the context of thermophoresis. We could also think of controlling the mechanical properties of the graphene sheets, maybe by deposition of graphene on patterned surface, by patterning graphene chemically or by functional groups.

These situations are very interesting but hypothetical up to now and remain to be explored. These are leads for the future.

■Comment 3.2

A further problem, is there any influence of the anisotropic structure of graphene on the driplon? What's the influence? i.e., armchair direction or zigzag direction.

●Reply 3.2

As mentioned in the Comment 3.1 by the reviewer, the shape of the driplon is isotropic and the effect of the anisotropic nature of the graphene sheets (armchair and zigzag directions) have not been observed. In order to confirm this observation, we have calculated the bending rigidity, a key parameter in driplon dynamics and thermodynamics, in the different directions. We obtained $B = 2.2$ eV for the bending along the zigzag direction, and $B = 2.1$ eV along the armchair direction. This is consistent with the fact that the effect of anisotropy is small. These values are mentioned **in the Supplemental Information Sec. S1 (highlighted in red color)**.

■Comment 3.3

The fastest growing mode is estimated to have a wavelength of 5.7nm, what is the meaning of the equilibrium wavelength should be 5.7nm?

●Reply 3.3

Actually the mode with this wavelength is the one which destabilizes the fastest the layer during the spinodal decomposition process (during a time scale typically \sim tens ps). This leads to a pattern – as shown in Fig. 3a and 3b, also SI Fig. S3 – which is mainly characterized by modes associated with the corresponding wavelength (here $\lambda_m = 5.7$ nm). Now as we show in the paper, this spinodal decomposition process leads to the formation of a driplon, which is the proper equilibrium state in the present condition. In this sense, this is not an equilibrium wavelength but that of the spinodal instability leading to the equilibrium state.

Note that this wavelength, which is estimated via scattering intensity function of the image (Fig.2b) is in very good agreement with the prediction of thin film theory (Eq. (2)), by means of the disjoining pressure curve in Fig. 2c.

■Comment 3.4

Why the diameter ($2R_s$) of the driplon at $t = 400ps$ is larger than the equilibrium wavelength 5.7nm?

●Reply 3.4

As in the above answer in Reply 3.3, the wavelength corresponds to the characteristic length of the pattern during the spinodal decomposition process. This corresponds to the early stage of the process, which in the longer time scale, leads to the formation of the driplon. In the longtime behavior at $t = 400$ ps is described rather in terms of the thermodynamics, see **the section “Thermodynamics of driplon creation”**. The equilibrium size can be larger than the unstable mode (since the two explore different region in the thermodynamic phase space, as highlighted by the corresponding free energy). These two stages are generic within the general dynamics of first order phase transitions.

■Comment 3.5

In the main text, the authors sometimes call these extra water molecules defects, sometimes call them driplon. I cannot find the definition of the defect through the manuscript. Are these terms different?

●Reply 3.5

We used the term “defects” when we wanted to emphasize the property of the driplon as a localized ripple in graphene sheets. However following the referee’s comment, we noticed that this term was not very clear at some places and we rephrased using other (defined) terms **in the abstract, and lines 519 and 530**.

■Comment 3.6

The formation of monolayer or bilayer is described by phase separation process. As long as the water density along thickness direction is inhomogeneous, such a phenomenon (FIG. 3) should widely exist. Why does the phase separation disappear, as shown in FIG. 2, when the number density ρ is large enough? i.e., above 23.9 nm^{-2} .

●Reply 3.6

As shown in Fig. 2b, when the surface density ρ is rather small and in the range $< 24 \text{ nm}^{-2}$, the water equilibrium state takes the form of mono- or bi-layer state, but intermediate confinement, for example 1.5-layer, cause phase separation. When ρ is large enough, however, the water can take disordered state and easily takes the continuous value of the height h , e.g. 2.5-layer state. This is quantitatively demonstrated in Fig. 4b, i.e., the phase separation occurs when the density ρ is between two wells, but does not happen when it is larger than second well.

■Comment 3.7

There are many mistakes in the manuscript. For example, the density in line 104 should be 13.1 nm^{-2} instead of 1.31 nm^{-2} . The authors should check the whole manuscript carefully.

●Reply 3.7

We thank the reviewer for pointing the typos. We have corrected this and other minor typos after careful checking.

■Comment 3.8

The potential applications of the so-called driplon should be discussed, otherwise, there is not much difference from playing with boundary value problem by molecular dynamic simulation.

●Reply 3.8

Actually we indeed believe that this driplon may deeply impact the dynamics and transport of water and fluids in layered graphene materials. Experiments along these line have just started in our group.

A discussion on several applications in the revised manuscript is proposed in the last paragraph (“conclusions and perspectives”). For example, fast particle transport is a possible application once the motion of driplon is controlled as discussed in Reply 3.1. The ripples that move quickly could act as cargoes transporting particles trapped between the sheets. The driplon structures are more stable when they contains particles inside thanks to the osmotic effect, as discussed in the context of Ostwald ripening. Therefore this

application is all the more promising, and the fast dynamics of the objects could allow to boost the transport of particles across these layered materials by adding new degrees of freedom for the strongly confined solute. From a more general perspective, these fast defects should impact water and ionic transport across such layered materials, as well as salt rejection. Such behavior would pave the way to possible applications including manipulation solutes via applied electric fields or osmotic gradients. These applications are discussed **in the last two paragraphs starting from line 517**.

■ **Comment 3.9**

I cannot find the citation of the relevant work in the references [Sci Rep. 2017; 7: 2646; J. Am. Chem. Soc., 2008, 130 (6), pp 1871–1878; Chin. Phys. B Vol. 26, No. 10 (2017) 106401; J. Phys. Chem. C 2007, 111, 1709-1715, etc], the difference from previous work should be explained and emphasized.

● **Reply 3.9**

We thank the referee for sharing the relevant literature. In the revised manuscript, we cite these papers **as Refs. 24, 25, 26, and 39**. Indeed, the fact that there exists this many relevant papers directly shows that the question of localized graphene structure has raised strong interests recently.

The present study focuses on the coupling between the localizing water and the flexible graphene sheets, in contrast to the references [J. Am. Chem. Soc. 2008, Chin. Phys. B 2017, J. Phys. Chem. C 2007] in which the water confined in the rigid graphene sheets are considered. The paper [Sci. Rep. 2017] considers the situation of a transition from a water-encapsulated structure to to a no-water graphene-graphene contact. In contrast, we consider the case of two phases, one is droplets with bilayer water, and the other is *water backgrounds* with the monolayer water state. It is highly surprising that the remaining water allows for localized structure, and this is (again) due to the disjoining contributions. This makes our driplon defect relevant for lamellar GO like structures, which are highly spread. The difference from the present study is mentioned **in the introduction around line 58**.

REVIEWERS' COMMENTS:

Reviewer #1 (Remarks to the Author):

In this revised manuscript, the authors have addressed all my previous issues. I recommend publication of this manuscript on Nature Communications.

Reviewer #2 (Remarks to the Author):

The authors have addressed all my comments. I am satisfied with the changes and the new version of the manuscript and I therefore suggest publication.

Reviewer #3 (Remarks to the Author):

All of my comments and suggestions are well addressed. It is my pleasure to recommend it for publication now.